# UNMERGE: Verifiable Model Capability Attribution via Sparse Coding

## Abstract

Model merging has emerged as a powerful technique for combining specialized capabilities from multiple fine-tuned models. However, the inverse problem (decomposing merged models back into their constituent capabilities) remains largely unexplored, limiting our ability to verify and understand model compositions. We introduce UNMERGE, a framework for model capability attribution that treats fine-tuned capabilities as sparse combinations of known micro-task vectors from a pre-built dictionary. Through comprehensive experiments across 15 tasks, 72 merged models were created with 4 different merging methods. Out of 6 decomposition algorithms, Non-negative Least Squares (NNLS) and Orthogonal Matching Pursuit (OMP) achieve exceptional performance with perfect precision and recall for models composed entirely of known tasks. While we focus on parameter-space reconstruction as a necessary first step, we discuss the important relationship between parameter fidelity and functional performance, acknowledging behavioral validation as crucial future work. Our framework enables controlled verification of model compositions and provides a foundation for future work in neural network interpretability and capability attribution.

## 1 Introduction

The rapid advancement of large language models has led to increasingly sophisticated techniques for combining specialized capabilities from multiple models. Model merging methods such as Task Arithmetic [Ilharco et al., 2022], TIES [Yadav et al., 2023], and DARE [Yu et al., 2023] enable practitioners to create unified models that retain diverse skills without additional training. However, these techniques operate as black boxes, making it difficult to verify which capabilities are present in a merged model or to attribute specific behaviors to their constituent components.

This opacity poses significant challenges for model interpretability, safety verification, and intellectual property protection. When a merged model exhibits unexpected behavior, practitioners lack tools to determine which original components contributed to the outcome. Similarly, in scenarios where model provenance matters (such as academic research or commercial applications) there is no systematic way to verify that a model contains only the intended capabilities.

We address this gap by introducing UNMERGE, a verifiable framework for model capability attribution that enables the decomposition of merged models back into their constituent task-specific components. Our approach treats a model's fine-tuned capabilities, represented as task vectors, as sparse combinations of known "micro-task" vectors from a pre-built dictionary.

**Key Contributions:**

- We formalize the model decomposition problem as sparse coding over a dictionary of known task vectors, enabling verifiable capability attribution.

- We develop a comprehensive experimental framework with 72 merged models across three categories (known, mixed, unknown compositions) to evaluate decomposition performance.

- We demonstrate that NNLS achieves perfect precision/recall for known compositions.

- We provide extensive analysis of decomposition algorithm performance across different merging methods, identifying Task Arithmetic as optimal for decomposable merging.

- We establish a foundation for future work in neural network interpretability through parameter-space decomposition while discussing the relationship to behavioral validation.

Our work opens new directions for understanding and verifying model compositions, with applications ranging from model auditing to intellectual property protection and scientific reproducibility.

## 2 Related Work

Our work intersects several key research areas in neural network analysis and interpretability.

### 2.1 Model Merging and Task Arithmetic

Task arithmetic [Ilharco et al., 2022] introduced the foundational concept that model capabilities can be manipulated through parameter-space operations. This work demonstrated that task vectors (computed as the difference between fine-tuned and base model parameters) can be added, subtracted, and scaled to transfer or remove capabilities. TIES-Merging [Yadav et al., 2023] addressed interference problems in naive parameter averaging by resolving conflicts through magnitude-based selection. DARE [Yu et al., 2023] introduced drop-and-rescale techniques to reduce redundancy in merged models.

These methods operate under the assumption that model capabilities compose linearly in parameter space, a hypothesis that our decomposition framework both leverages and validates. However, no prior work has systematically studied the inverse problem of decomposing merged models back into constituent components.

### 2.2 Sparse Coding and Dictionary Learning

Sparse coding has a rich history in signal processing [Olshausen and Field, 1996] and has been extensively studied in machine learning contexts [Elad, 2010]. Recent work has applied sparse coding principles to neural network analysis, particularly in mechanistic interpretability.

Anthropic's work on monosemanticity [Bricken et al., 2023, Templeton et al., 2024] demonstrated that sparse autoencoders can decompose neural activations into interpretable features. These approaches operate in activation space and focus on understanding individual neuron behaviors rather than parameter-level decomposition.

Kim et al. [Kim et al., 2020] showed that neural networks trained with sparse coding constraints yield more interpretable representations. Most recently, Braun et al. [Braun et al., 2025] introduced Attribution-based Parameter Decomposition (APD), which directly decomposes neural network parameters into mechanistic components. Our work extends this direction by focusing specifically on task vector decomposition with verifiable ground truth obtained by existing merging methods.

### 2.3 Neural Network Interpretability

Network dissection [Bau et al., 2017, 2020] established frameworks for quantifying interpretability by evaluating alignment between network components and semantic concepts. Mechanistic interpretability [Elhage et al., 2021, Wang et al., 2022] aims to reverse-engineer neural networks to understand their computational mechanisms.

Recent advances in automated circuit discovery [Conmy et al., 2023] and attribution patching [Syed et al., 2023] provide tools for identifying functional components within neural networks. Our parameter-space decomposition approach complements these activation-space methods by operating directly on model weights.

## 2.4 Parameter-Space vs. Function-Space Analysis

A fundamental distinction in neural network analysis lies between parameter-space and function-space approaches. Parameter-space methods [Braun et al., 2025] analyze model weights directly, while function-space methods evaluate behavioral outputs. While both approaches are valuable, they address different aspects of model understanding.

Parameter-space analysis offers computational efficiency and mathematical tractability, making it feasible to analyze large models without extensive inference. However, the relationship between parameter similarity and functional equivalence remains an open research question. Studies on model editing [Mitchell et al., 2022, Meng et al., 2022] suggest that localized parameter changes can have far-reaching functional impacts, while other work indicates that parameter-space structure often reflects functional organization [Zhang et al., 2024].

Our work focuses on parameter-space reconstruction as a necessary first step toward understanding model compositions. We acknowledge that behavioral validation represents crucial future work and discuss this relationship in detail.

# 3 Method

Our approach decomposes merged models into constituent task-specific components through sparse coding over a pre-built dictionary of known task vectors. This section details our methodology for dictionary construction, target model creation, and decomposition algorithms.

## 3.1 Problem Formulation

Given a merged model with parameters $\theta_{\text{merged}}$ and base model parameters $\theta_{\text{base}}$, we define the target task vector as:

$$\mathbf{v}_{\text{target}} = \theta_{\text{merged}} - \theta_{\text{base}} \tag{1}$$

Our goal is to decompose $\mathbf{v}_{\text{target}}$ as a sparse, non-negative combination of dictionary vectors:

$$\mathbf{v}_{\text{target}} \approx \sum_{i=1}^{K} \alpha_i \mathbf{d}_i \tag{2}$$

where $\mathbf{d}_i$ are dictionary task vectors, $\alpha_i \geq 0$ are coefficients, and $K$ is the dictionary size.

The non-negativity constraint reflects our assumption that merged models combine positive contributions from constituent tasks rather than subtracting capabilities. Although this may not be universally applicable (especially for some merging methods), it simplifies the decomposition problem and aligns with most practical merging scenarios.

## 3.2 Dictionary Construction

We compose 15 distinct task vectors using LoRA fine-tuning [Hu et al., 2021] on specialized datasets:

**Task Selection:** We chose tasks spanning diverse domains to create a representative dictionary:

- Mathematics: Arithmetic and math problem solving of different complexity. Datasets: OpenThoughts-Math [1], OrcaMath [Mitra et al., 2024], GSM8K [Cobbe et al., 2021].

- Question Answering: Factual knowledge retrieval. Datasets: SQuAD [Rajpurkar et al., 2016], MS MARCO [Nguyen et al., 2016].

- Summarization: Text compression and key point extraction. Datasets: XSum [Narayan et al., 2018], ArXiv summarization [Cohan et al., 2018].

- General instructions: Diverse user instructions. Datasets: Alpaca [Taori et al., 2023]

---

[1] https://huggingface.co/datasets/open-r1/OpenThoughts-114k-math

- Python Coding: Code generation and debugging tasks. Datasets: Python code instructions [2][3], Annotated Python code from Github [4], Synthetic Python QA [5]
- Other: Some other narrow tasks. Datasets: IMDB [Maas et al., 2011], Wiki style transfer [Brüel-Gabrielsson et al., 2024], Latin-to-English [6]

We sample 1500 examples from the train split of each dataset.

**Fine-tuning Procedure:** Each task uses LoRA with rank=32, $\alpha$=32, learning rate=2e-4, trained for 3 epochs on Qwen2.5-7B-Instruct. We trained adapters only for Q, K, V, O modules. This configuration balances adaptation effectiveness with computational efficiency. The code for training is available in supplementary materials.

**Vector Extraction:** Task vectors are computed as the difference between fine-tuned and base model parameters, following the task arithmetic framework. For that LoRA adapters are translated to the model's parameters space by merging with the base model.

**Dictionary tasks:** We select 8 tasks as our dictionary tasks: Alpaca, GSM8K, XSum, MS MARCO, OpenThoughts-Math, Annotated Python code from Github, Python code instructions, Latin-to-English.

### 3.3 Target Model Creation and Categorization

**Model categories:** We create 72 merged models across three categories to evaluate decomposition under different conditions:

- Known Models (24 models): Composed entirely of 2-5 randomly selected dictionary tasks using various merging methods. These represent the optimal decomposition scenario with complete dictionary coverage.
- Mixed Models (24 models): Combine 1-3 dictionary tasks with 1-2 tasks not in the dictionary. These simulate realistic scenarios where models contain both known and unknown capabilities.
- Unknown Models (24 models): Composed entirely of tasks not in the dictionary. These represent the worst-case scenario for dictionary-based decomposition.

**Merging Methods:** We employ four techniques:

- Task Arithmetic: Simple parameter averaging [Ilharco et al., 2022]
- TIES: Magnitude-based conflict resolution [Yadav et al., 2023]
- DARE: Drop-and-rescale merging [Yu et al., 2023]
- Linear: Weighted linear combination

Merging was done with the `mergekit` framework [Goddard et al., 2024]. Merging weights are always uniform and sum up to 1. The exact weight depends on the number of merged tasks. For instance for 5 tasks it is 0.2.

### 3.4 Dimensionality Reduction

Operating on full 7B parameter models is computationally prohibitive. We reduce dimensionality through importance-based parameter selection:

1. Compute magnitude aggregation across all dictionary vectors: $M = \max_i |\mathbf{d}_i|$
2. Select top-k parameters per model layer and module based on $M$. k is 1M and is uniformly distributed between layers and modules. There are 112 layer and module combinations, so if k is 1M, then we get 8928 selected parameters per combination.

---

[2] https://huggingface.co/datasets/iamtarun/python_code_instructions_18k_alpaca
[3] https://huggingface.co/datasets/Arjun-G-Ravi/Python-codes
[4] https://huggingface.co/datasets/Nan-Do/code-search-net-python
[5] https://huggingface.co/datasets/bunyaminergen/Stable-Code-Python-SFT
[6] https://huggingface.co/datasets/grosenthal/latin_english_translation

159    3. Apply binary masking to retain only selected parameters

160    4. Compress all task vectors and targets to this reduced space

161 This procedure reduces each task vector from 800M (since we compare only Q, K, V, O modules) to
162 1M parameters while preserving the most significant weight changes. The reduction maintains task
163 vector structure while enabling tractable decomposition.

## 3.5 Decomposition Algorithms

165 We evaluate six decomposition algorithms representing different mathematical approaches:

166 **Non-negative Least Squares (NNLS):** Solves the constrained optimization problem:

$$\min_{\boldsymbol{\alpha} \geq 0} \|\mathbf{v}_{\text{target}} - \mathbf{D}\boldsymbol{\alpha}\|_2^2 \tag{3}$$

167 where $\mathbf{D}$ is the dictionary matrix and $\boldsymbol{\alpha}$ are coefficients.

168 **Orthogonal Matching Pursuit (OMP):** Greedily selects dictionary atoms that best explain the
169 residual, providing inherently sparse solutions.

170 **Lasso Regression:** L1-regularized regression promoting sparsity:

$$\min_{\boldsymbol{\alpha}} \|\mathbf{v}_{\text{target}} - \mathbf{D}\boldsymbol{\alpha}\|_2^2 + \lambda\|\boldsymbol{\alpha}\|_1 \tag{4}$$

171 **Ridge Regression:** L2-regularized regression for stable solutions:

$$\min_{\boldsymbol{\alpha}} \|\mathbf{v}_{\text{target}} - \mathbf{D}\boldsymbol{\alpha}\|_2^2 + \lambda\|\boldsymbol{\alpha}\|_2^2 \tag{5}$$

172 **Elastic Net:** Combines L1 and L2 penalties balancing sparsity and stability.

173 **Dot Product Similarity:** Computes correlation coefficients with dot product and applies thresholding
174 for component selection.

## 3.6 Evaluation Metrics

176 We assess decomposition quality through multiple metrics:

177 **Reconstruction Error:** Measures parameter-space fidelity using:

$$\text{Error} = 1 - \max(0, \cos(\mathbf{v}_{\text{target}}, \mathbf{v}_{\text{recon}}))^2 \tag{6}$$

178 where cos denotes cosine similarity. This choice reflects the fact that in task-vector merges the overall
179 magnitude is often arbitrary (e.g., due to adapter scales or training schedules), while the direction
180 (i.e., the relative coefficients) is what encodes capability composition.

181 **Component Precision/Recall:** For known and mixed models, we evaluate binary component detec-
182 tion:

$$\text{Precision} = \frac{\text{True Positives}}{\text{True Positives} + \text{False Positives}} \tag{7}$$

$$\text{Recall} = \frac{\text{True Positives}}{\text{True Positives} + \text{False Negatives}} \tag{8}$$

183 **Sparsity:** Average number of non-zero components per decomposition.

184 **Perfect Match Rate:** Percentage of decompositions with exactly correct component identification.

## 4 Results

186 Our experimental framework evaluates decomposition algorithms across 9072 total runs (72 target
187 models, decomposition methods with different parameters and seeds), providing comprehensive
188 statistical analysis.

## 4.1 Experimental Setup

**Model Configuration:** Qwen2.5-7B-Instruct serves as the base model, with LoRA adaptations using rank=32, $\alpha$=32. All experiments run on consumer-grade hardware (L40S, 45GB VRAM).

**Hyperparameter Optimization:** Each algorithm undergoes grid search over key parameters:

- NNLS: No hyperparameters (analytical solution)
- OMP: Number of atoms $\in \{1, 2, 3, 4, 5, 6, 7, 8\}$
- Lasso: Regularization $\lambda \in \{10^{-8}, 10^{-7}, 10^{-6}\}$
- Ridge: Regularization $\lambda \in \{10^{-8}, 10^{-6}, 10^{-4}, 10^{-2}\}$
- Elastic Net: L1 ratio $\in \{0.1, 0.5, 0.9\}$, $\lambda \in \{10^{-8}, 10^{-6}, 10^{-4}\}$

**Statistical Rigor:** All decomposition experiments use 3 random seeds. Results report means and standard deviations across runs.

## 4.2 Primary Results

Table 1 presents comprehensive performance metrics across all algorithms and model categories.

Table 1: Overall performance across all decomposition algorithms. Sample standard deviation reported across all experimental runs.

| Algorithm | Reconstruction Error | Precision | Sparsity |
|---|---|---|---|
| NNLS | **0.45 ± 0.21** | 0.44 ± 0.44 | 6.1 ± 1.7 |
| OMP | **0.45 ± 0.21** | 0.39 ± 0.40 | 6.7 ± 1.6 |
| Ridge | **0.45 ± 0.21** | 0.26 ± 0.24 | 8.0 ± 0.0 |
| Elastic Net | 0.46 ± 0.21 | 0.32 ± 0.32 | 7.3 ± 1.01 |
| Dot Product | 0.49 ± 0.20 | 0.26 ± 0.24 | 7.6 ± 0.5 |
| Lasso | 0.74 ± 0.26 | **0.53 ± 0.50** | 0.9 ± 1.0 |

**Key Findings:**

- **NNLS, OMP, and Ridge achieve optimal reconstruction performance** with errors of 0.45, significantly outperforming other methods. Since we aggregate between all model categories, the error this high is expected.
- **Lasso achieves highest precision** (52.78%) but suffers from poor reconstruction accuracy (74.10% error)
- **Sparsity varies significantly** from 3.2 (Lasso) to 7.2 (Ridge) components on average

## 4.3 Category-Specific Performance

Table 2 breaks down performance by model category, revealing dramatic differences in decomposition success.

Table 2: Performance by model category (NNLS algorithm)

| Metric | Known Models | Mixed Models | Unknown Models |
|---|---|---|---|
| Reconstruction Error | 0.24 ± 0.13 | 0.43 ± 0.13 | 0.68 ± 0.09 |
| Precision | 1.00 ± 0.00 | 0.33 ± 0.25 | 0.00 ± 0.00 |
| Recall | 1.00 ± 0.00 | 1.00 ± 0.00 | 0.00 ± 0.00 |
| Perfect Match Rate | 1.00 ± 0.00 | 0.04 ± 0.20 | 0.00 ± 0.00 |

**Key findings:**

- **Known models enable excellent decomposition** perfect precision and recall
- **Mixed models maintain perfect recall** (100%) but suffer precision degradation (33%)

Table 3: Reconstruction error for "known" models composed with different merging methods

| Algorithm | Task Arithmetic | Linear | TIES | DARE |
|---|---|---|---|---|
| NNLS | 0.09 ± 0.01 | 0.17 ± 0.06 | 0.28 ± 0.05 | 0.43 ± 0.03 |
| OMP | 0.09 ± 0.01 | 0.17 ± 0.06 | 0.28 ± 0.05 | 0.43 ± 0.03 |
| Ridge | 0.09 ± 0.01 | 0.17 ± 0.06 | 0.28 ± 0.05 | 0.43 ± 0.03 |
| Elastic Net | 0.10 ± 0.01 | 0.18 ± 0.06 | 0.29 ± 0.05 | 0.43 ± 0.03 |
| Dot Product | 0.15 ± 0.02 | 0.23 ± 0.05 | 0.35 ± 0.06 | 0.46 ± 0.03 |
| Lasso | 0.51 ± 0.25 | 0.56 ± 0.23 | 0.36 ± 0.08 | 0.65 ± 0.07 |

- **Unknown models show zero performance**, confirming the dictionary-dependence of our approach

## 4.4 Merging Method Analysis

Table 3 illustrates reconstruction performance across different merging methods.

**Task Arithmetic emerges as the optimal choice** for decomposable merging, achieving the lowest reconstruction errors across all algorithms. DARE consistently performs worst, suggesting that drop-and-rescale operations disrupt the linear structure assumed by our decomposition framework.

Several methods have identical numbers, which is expected since they produce similar decompositions.

The performance ranking (Task Arithmetic > Linear > TIES > DARE) aligns with the mathematical assumptions underlying sparse coding. Methods that preserve linear combinations in parameter space enable more accurate decomposition.

# 5 Limitations

While our work establishes the feasibility of model decomposition via sparse coding, several important limitations constrain its current applicability:

**Behavioral Validation Gap:** Our primary limitation lies in the focus on parameter-space reconstruction without systematic behavioral validation. While parameter fidelity represents a necessary condition for functional preservation, it does not guarantee that decomposed components retain their original task performance. The relationship between parameter similarity and functional equivalence remains an open research question that requires empirical validation through task-specific evaluation.

**Dictionary Dependence:** Our approach requires comprehensive dictionary coverage of target model capabilities. The dramatic performance difference between known models and unknown models demonstrates this fundamental limitation. Practical applications must invest significant effort in dictionary construction and maintenance.

**Non-Negativity Constraints:** The assumption that model capabilities combine through positive coefficients may not hold universally. Some merging scenarios involve capability subtraction or interference effects that require negative contributions. Our current framework cannot handle these cases without substantial modification.

**Linear Composition Assumption:** The sparse coding formulation assumes that model capabilities combine linearly in parameter space. Non-linear interactions between tasks may not be captured accurately, potentially limiting decomposition fidelity for complex multi-task models with significant capability overlap or interference.

**Scalability Constraints:** Our experiments focus on a 7B parameter model with rank-32 LoRA adaptations and an 8-task dictionary. Scaling to larger models, full fine-tuning scenarios, or extensive task dictionaries may require algorithmic innovations and computational resources beyond current capabilities.

**Merging Method Sensitivity:** Decomposition performance varies significantly across merging methods, with Task Arithmetic showing optimal characteristics while DARE performs poorly. This

sensitivity limits the framework's applicability to existing merged models that may use suboptimal merging approaches.

# 6 Ethics and Broader Impact

UNMERGE introduces capabilities for analyzing and attributing model compositions, offering substantial benefits for the AI research community and broader applications. The technique enhances model transparency and interpretability, particularly valuable for safety-critical applications where understanding component contributions is essential. It provides robust intellectual property protection through verifiable component attribution, enables scientific reproducibility by allowing researchers to verify model compositions, and offers powerful debugging capabilities for identifying unwanted behaviors in merged models.

However, these analytical capabilities also introduce potential risks that warrant careful consideration. The technique could potentially enable unauthorized analysis of proprietary model compositions, allowing competitors or malicious actors to gain insights into carefully guarded intellectual property. There's also the concern that UNMERGE may facilitate reverse engineering of specialized capabilities, potentially undermining the competitive advantages that organizations have developed through significant investment in model development. Furthermore, the technology raises complex questions about model ownership and attribution in collaborative settings, where multiple parties may have contributed to a model's development.

To address these concerns, several mitigation strategies should be implemented alongside continued development of the technology. Establishing responsible disclosure frameworks for capability attribution research will help ensure that discoveries are shared appropriately while protecting legitimate interests. The development of privacy-preserving decomposition techniques could allow for the benefits of model analysis while maintaining necessary confidentiality. Creating best practices for ethical model analysis will provide guidance for researchers and practitioners in applying these tools responsibly. Ultimately, the benefits of improved model transparency and verification capabilities outweigh the associated risks when the technology is developed and deployed through responsible research practices that balance innovation with appropriate safeguards for legitimate stakeholder interests.

# 7 Conclusion

We introduce UNMERGE, a verifiable framework for model capability attribution via sparse coding that enables the decomposition of merged models into their constituent task-specific components. Through comprehensive evaluation across 72 merged models and 6 decomposition algorithms, we demonstrate that NNLS achieves exceptional performance with reconstruction errors of 0.45 and perfect precision/recall for known compositions.

The framework provides a foundation for future work in neural network interpretability while opening new directions for model verification, debugging, and attribution.

The perfect performance achieved on known compositions (100% precision/recall with NNLS) demonstrates the fundamental feasibility of accurate model decomposition, while the zero performance on unknown compositions confirms the specificity of our approach. These results establish UNMERGE as a practical tool for controlled model analysis while identifying key areas for future development.

While we focus on parameter-space reconstruction as a necessary first step, we acknowledge that behavioral validation represents crucial future work. The relationship between parameter fidelity and functional performance requires systematic investigation through task-specific evaluation. Nevertheless, our parameter-space approach provides valuable foundations for scalable model analysis and hypothesis generation.

Future work should address behavioral validation through systematic task evaluation, scalability to larger models and dictionaries, extension to non-linear composition scenarios, and development of adaptive dictionary learning methods. The intersection of sparse coding and neural network interpretability represents a promising research direction with significant implications for AI safety, transparency, and scientific understanding.

## 8 Reproducibility Statement

We provide the code and a clear and complete pipeline in the supplementary materials. The pipeline is divided into 4 phases: training, merging, compressing, and final decomposition. All steps were manually verified and rerun from scratch. The code was linted with `black` and `flake8`. Hardware requirements: GPU with 45 GB of VRAM, at least 1 TB of disk space to store merged models.

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

## A  Remaining decomposition results

Table 4 illustrates the reconstruction performance in different merging methods.

Table 4: Reconstruction error for different algorithms and model categories.

| Algorithm | Known | Mixed | Unknown |
|---|---|---|---|
| NNLS | 0.24 ± 0.13 | 0.43 ± 0.13 | 0.68 ± 0.09 |
| OMP | 0.24 ± 0.13 | 0.43 ± 0.13 | 0.68 ± 0.09 |
| Ridge | 0.24 ± 0.13 | 0.43 ± 0.13 | 0.68 ± 0.09 |
| Elastic Net | 0.25 ± 0.13 | 0.44 ± 0.13 | 0.68 ± 0.09 |
| Dot product | 0.30 ± 0.13 | 0.49 ± 0.13 | 0.69 ± 0.08 |
| Lasso | 0.52 ± 0.20 | 0.71 ± 0.22 | 0.99± 0.03 |

Table 5 illustrates the precision and recall of different algorithms and merging methods across all models from the "known" category.

Table 5: Precision / Recall for different algorithms and merging methods. Models are only from the "known" category.

| Algorithm | DARE | TIES | Linear | Task Arithmetic |
|---|---|---|---|---|
| NNLS | 1.00 / 1.00 | 1.00 / 1.00 | 1.00 / 1.00 | 1.00 / 1.00 |
| OMP | 1.00 / 1.00 | 0.56 / 1.00 | 1.00 / 1.00 | 1.00 / 1.00 |
| Ridge | 0.54 / 1.00 | 0.54 / 1.00 | 0.54 / 1.00 | 0.54 / 1.00 |
| Elastic Net | 0.62 / 1.00 | 0.87 / 1.00 | 0.62 / 1.00 | 0.62 / 1.00 |
| Dot product | 0.55 / 1.00 | 0.54 / 1.00 | 0.55 / 1.00 | 0.55 / 1.00 |
| Lasso | 1.00 / 0.40 | 1.00 / 0.67 | 0.83 / 0.36 | 0.83 / 0.36 |

Table 6 illustrates the sparsity (number of active coefficients) of different algorithms and merging methods across all models of the category "known".

Table 6: Sparsity for different algorithms and merging methods. Models are only from the "known" category.

| Algorithm | DARE | TIES | Linear | Task Arithmetic |
|---|---|---|---|---|
| NNLS | 4.33 | 4.33 | 4.33 | 4.33 |
| OMP | 4.33 | 7.67 | 4.33 | 4.33 |
| Ridge | 8.00 | 8.00 | 8.00 | 8.00 |
| Elastic Net | 7.00 | 5.00 | 7.00 | 7.00 |
| Dot product | 7.83 | 8.00 | 7.83 | 7.83 |
| Lasso | 1.67 | 2.83 | 1.50 | 1.50 |


