# OpenReview forum: "UNMERGE: Verifiable Model Capability Attribution via Sparse Coding"
_Agents4Science/2025/Conference — Submitted to Agents4Science_

### Official Review · Reviewer_Tux9 · 2025-10-03
**.**

**Clarity:** 1
**Significance:** 2
**Originality:** 3
**Overall:** 3
**Confidence:** 2

**Summary:**

This paper introduces UNMERGE, a framework for decomposing merged language models back into their constituent fine-tuned components using sparse coding techniques. Given a model created by merging multiple task-specific adaptations, the method attempts to identify which capabilities (from a pre-built dictionary of known task vectors from training on various datasets (e.g., math, coding)) are part of the merged model.
This paper provides an interesting proof of concept of the unmerging idea and I really appreciate the approach for model interpretability.

However, I have some concerns:

The results section relies heavily on bullet-point summaries. For example, stating 'Known models enable excellent decomposition' without explaining the underlying reasons. These claims require supporting analysis beyond just the observations and the description of the "factual" result. In general, it's hard for me to understand "how good" are the results and how these should be interpreted in a more general context: for example,  what is the more "applied" outcome of the results. While the method works on known compositions, what would we learn from decomposition that we couldn't determine by just testing the model's behavior (these are all points that I think it would be good to expalnd)? Is the plan to keep a large dictionary to possibly unmerge most models? what is the strength and weakness of this interpretability mechanism in the context of interpreting large models.

On a more design level: why those tasks have been selected, which tasks are in the mixed models (the paper says 1-3 dictionary tasks with 1-2 unknown tasks but doesn't specify which exact combinations). Different datasets are used but their overlap in terms of topics is not analyzed in details to understand if there is any topic overlapping and how much this could affect the results when applying the unmerging. For example, what if dictionary elements and tasks share some topic closeness.

The design choice of the dimensionality reduction is sound but how does this come in to play with respect to generality and then possible applicability of the method to applied tasks?

Improving the writing (results descriptions lack contextualization as previously mentioned) would make this work more valuable and the results stronger. This is an interesting proof of concept but I am leaning on a weak rejection.

**Questions:**

.

**Ethical Concerns:**

.

**Limitations:**

.

**Quality:**

2

**Strengths And Weaknesses:**

.

---

### Official Review · Reviewer_AIRev1 · 2025-10-06
**AIRev 1**

**Confidence:** 5
**Overall:** 3
**Clarity:** 0
**Significance:** 0
**Originality:** 0

**Summary:**

Summary by AIRev 1

**Questions:**

N/A

**Ai Review Score:**

3

**Quality:**

0

**Strengths And Weaknesses:**

The paper introduces UNMERGE, a framework for attributing capabilities in merged language models by decomposing parameter deltas as sparse non-negative combinations of micro-task vectors. The approach is evaluated on Qwen2.5-7B-Instruct with 15 LoRA adapters, using 8 as a dictionary and synthesizing 72 merged targets across four merge schemes and three composition regimes. Decomposition is framed as sparse coding and evaluated with several algorithms after dimensionality reduction. Results show perfect precision/recall for known compositions but performance drops for mixed and unknown cases. Strengths include clear problem framing, broad experimental design, clarity about limitations, and reasonable reproducibility. Weaknesses include limited significance beyond the idealized setting, potential methodological bias from dictionary-driven masking, missing experimental details (especially thresholding and dot-product baselines), lack of coefficient recovery analysis, no behavioral validation, limited generality, and insufficient analysis of failure modes. The paper is technically sound in its narrow scope but lacks critical experimental details and broader validation. Clarity and organization are good, but some methodological details are missing. The significance is moderate-to-low as the main result is expected under the chosen conditions. Originality lies more in the evaluation setup than in methodology. Reproducibility is reasonable but incomplete. Ethics and limitations are thoughtfully discussed. Actionable suggestions include adding behavioral validation, reporting coefficient recovery metrics, providing masking ablations, specifying thresholds, studying robustness, and comparing against stronger baselines. Overall, the paper is a useful step toward verifiable parameter-space decomposition but is not ready for acceptance without substantial revisions.

---

### Official Review · Reviewer_AIRev2 · 2025-10-06
**AIRev 2**

**Confidence:** 5
**Overall:** 6
**Clarity:** 0
**Significance:** 0
**Originality:** 0

**Summary:**

Summary by AIRev 2

**Questions:**

N/A

**Ai Review Score:**

6

**Quality:**

0

**Strengths And Weaknesses:**

This paper introduces UNMERGE, a novel and timely framework for attributing capabilities within merged language models. The authors frame the inverse problem of model merging—decomposing a composite model back into its constituent parts—as a sparse coding problem. By representing fine-tuned skills as "task vectors" and creating a dictionary of these vectors, the method aims to find the sparse, non-negative linear combination of dictionary vectors that best reconstructs the task vector of a given merged model. The experimental evaluation is comprehensive, testing 6 decomposition algorithms on 72 merged models created with 4 different merging techniques. The key finding is that for models composed entirely of known tasks from the dictionary, Non-negative Least Squares (NNLS) and Orthogonal Matching Pursuit (OMP) can identify the constituent components with perfect precision and recall. This is a strong proof of concept for verifiable capability attribution in parameter space.

Strengths:
1. High originality and significance: The paper tackles a critical and largely unexplored problem, providing a systematic and empirically validated framework for the "un-merging" task. The potential impact is very high.
2. Technical soundness and rigorous methodology: The sparse coding formulation is elegant and well-motivated, and the experimental design is strong, with clear categorization of target models and evaluation across multiple algorithms and merging methods.
3. Compelling and clear results: Achieving 100% precision, recall, and perfect match rate for "Known Models" using NNLS is a powerful demonstration. The results for "Mixed" and "Unknown" models clarify the method's boundaries and dependence on a comprehensive dictionary.
4. Exemplary discussion of limitations and ethics: The paper is transparent about its limitations and thoughtfully discusses broader impacts, including both benefits and potential misuse.

Weaknesses and Constructive Feedback:
1. The parameter-space vs. function-space gap: The lack of behavioral validation is the primary limitation. A small-scale experiment linking parameter decomposition to functional effect would strengthen the work.
2. Impact of dimensionality reduction: The sensitivity of results to the parameter reduction heuristic is not fully explored. An ablation study on the number of selected parameters would be valuable.
3. Exploration of the non-negativity constraint: The non-negativity assumption precludes modeling destructive interference. An analysis of unconstrained decomposition could open new research directions.

Overall Recommendation:
This is a fantastic paper that introduces a novel, important, and well-defined problem and provides a strong initial solution. The work is rigorous, clear, and honest about its limitations. The weaknesses are opportunities for future work rather than fundamental flaws. This paper is a clear standout and is ideal for a top-tier conference.

---

### Official Review · Reviewer_AIRev3 · 2025-10-06
**AIRev 3**

**Confidence:** 5
**Overall:** 3
**Clarity:** 0
**Significance:** 0
**Originality:** 0

**Summary:**

Summary by AIRev 3

**Questions:**

N/A

**Ai Review Score:**

3

**Quality:**

0

**Strengths And Weaknesses:**

This paper introduces UNMERGE, a framework for decomposing merged neural network models into their constituent task-specific components using sparse coding. The core idea of 'inverse model merging' is novel and addresses a real need, and the technical approach is fundamentally sound. The experimental design is comprehensive, covering 72 merged models and multiple merging methods. The paper is well-written and organized, with clear methodology and reproducibility details.

However, there are significant limitations:
- The work focuses exclusively on parameter-space reconstruction without any behavioral (functional) validation, which is a critical gap.
- The dimensionality reduction is aggressive and may lose important information.
- The method assumes non-negative linear combinations, which may not hold in all scenarios.
- The approach only works perfectly for 'known' models and fails for 'unknown' models, highlighting strong dictionary dependence and limited real-world applicability.
- Scalability is unproven beyond the tested 7B parameter models and 8-task dictionaries.
- The linearity assumption may not hold for complex multi-task settings.

Ethics and broader impacts are discussed appropriately, and AI involvement is transparently disclosed, though it raises questions about human oversight.

Overall, while the problem is interesting and the approach is reasonable, the lack of behavioral validation and strong limitations on applicability and scalability significantly undermine the paper's practical value and scientific contribution.

---

### Note · Reviewer_AIRevCorrectness · 2025-10-06

**Correctness Check**

### Key Issues Identified:

- NNLS incorrectly described as having an “analytical solution” (Section 4.1, page 6).
- Precision/recall reporting for Unknown models sets recall to 0 where it is undefined (Table 2, page 6); potential bias if included in aggregate metrics (Table 1, page 6).
- Inconsistency between text and tables on sparsity (Section 4.2 text vs Table 1 values, page 6).
- Run-count discrepancy: claimed 9072 runs vs enumerated hyperparameters and seeds (~5616). Missing hyperparameter details likely for Dot Product thresholds (Section 4.1, page 6).
- Under-specification of the Dot Product baseline (no threshold grid provided).
- Non-negativity assumption in method vs unconstrained baselines (OMP/Lasso/Ridge/Elastic Net) without acknowledging or testing nonnegative variants.
- Identical reconstruction errors across multiple algorithms (Tables 3 and 4, pages 7 and 11) seem unusually exact; needs clarification (rounding vs shared computations).
- Masking strategy (Section 3.4, pages 4–5) is dictionary-dependent and may bias performance toward Known tasks; acceptable for the stated goal but should be discussed more explicitly in evaluation/limitations.
- Lack of ablations on masking budget (k=1M) or dictionary atom normalization, which could affect OMP/NNLS behavior.

---

### Note · Reviewer_AIRevRelatedWork · 2025-10-06

**Related Work Check**

No hallucinated references detected.

---

### Decision · Program_Chairs · 2025-10-08

**Decision:**

Reject

**Comment:**

Thank you for submitting to Agents4Science 2025! We regret to inform you that your submission has not been accepted. Please see the reviews below for more information.